# Investigation of a Ventilation System for Energy Efficiency and Indoor Environmental Quality in a Renovated Historical Building: A Case Study

**DOI:** 10.3390/ijerph16214133

**Published:** 2019-10-27

**Authors:** Richard Nagy, Ľudmila Mečiarová, Silvia Vilčeková, Eva Krídlová Burdová, Danica Košičanová

**Affiliations:** 1Institute of Architectural Engineering, Faculty of Civil Engineering, Technical University of Košice, Vysokoškolská 4, 04200 Košice, Slovakia; richard.nagy@tuke.sk (R.N.); danica.kosicanova@tuke.sk (D.K.); 2Institute of Environmental Engineering, Faculty of Civil Engineering, Technical University of Kosice, Vysokoškolská 4, 04200 Košice, Slovakiaeva.kridlova.burdova@tuke.sk (E.K.B.)

**Keywords:** historical building, energy performance, natural ventilation, mechanical ventilation, IEQ, CO_2_, PM, SBS symptoms

## Abstract

This paper emphasizes the importance of environmental protection regarding the reduction of energy consumption while maintaining living standards. The aim of the research is to observe the effects of mechanical and natural ventilation on energy consumption and building operation as well as indoor environmental quality (IEQ). The results of indoor environmental quality testing show that the mean relative humidity (31%) is in the permissible range (30%–70%); the mean CO_2_ concentration (1050.5 ppm) is above the recommended value of 1000 ppm according to Pettenkofer; and the mean PM_10_ concentration (43.5 µg/m^3^) is under the limit value of 50 µg/m^3^. A very large positive correlation is found between relative humidity and concentration of CO_2_ as well as between the concentration of PM_5_ and the concentration of CO_2_. The most commonly occurring sick building syndrome (SBS) symptoms are found to be fatigue and the feeling of a heavy head.

## 1. Introduction

Heat is necessary for many building applications and, therefore, must be generated, stored, and used efficiently in terms of economic and environmental aspects [1]. As the building sector accounts for a large percentage of global energy consumption, it is essential to pay close attention to energy efficiency. Heating, ventilation, and air conditioning (HVAC) systems are largely responsible for the energy consumption of buildings. From HVAC systems, ventilation is a key issue in ensuring adequate indoor air quality (IAQ). Ventilation systems therefore play an important role not only in improving the energy efficiency of buildings, but also in ensuring a better indoor climate for users and reducing health problems [2]. In recent years, researchers have developed many energy efficient ventilation methods to reduce the energy consumption of buildings. Cuce et al. [3] showed that domestic buildings have the largest share with 63%, and most of the energy is utilized for HVAC systems in those buildings. This study also suggested reducing the energy consumption through waste heat recovery in HVAC systems. A study by Merzkirch et al. [4] showed that the energy efficient operation of mechanical ventilation systems depends on parameters such as the main air flow, internal and external recirculation, sensitivity to differential pressure, specific fan power, and heat recovery efficiency. In [5], Cuce and Riffat showed that heat recovery systems are able to recycle about 60%–95% of the waste energy, which is very promising. In dealing with an appropriate ventilation strategy, Guillén-Lambea et al. [6] stated that cooling is less energy-demanding than heating, except in hot climates, where a high cooling requirement makes it impossible to achieve a near-zero energy (nZEB) building without energy recovery. The heating and cooling requirements of nZEB without heat recovery can be met in mild climates. This study also found that the heating and cooling demand required by the Passivhaus can be achieved under a few warm climatic conditions. In other cases, it will be necessary to change the ventilation strategy and heat recovery ventilation. The numerical results presented in [7] indicate that passive cooling can simultaneously reduce average indoor temperatures and the unevenness of the internal thermal distribution. The subsequent energy performance analysis showed that the energy efficiency class for ventilation and cooling could be reduced by promoting the heat recovery efficiency of the ventilation equipment. In addition, the energy saving ratio of the air cooling unit decreased with increasing temperatures in exhaust air and heat flow for passive cooling in the concrete ceiling core. An experiment carried out by Khabbaz et al. [8] showed that the earth-to-air heat exchanger is a good semi-passive system for air refreshment, as the recorded blown air temperature into the building is quasi-constant at 25 °C with an air humidity level of around 40%, even though the outside temperature reaches more than 40 °C. In another study [9], a water loop was established between the exhaust air and the fresh air at which the heat pump was applied in the water loop to intensify the heat recovery process. One of the main results was that the ambient air temperature and the inlet water temperature of the heat recovery heat exchanger influenced the quantity of heat recovery. This study highlighted that the lower the ambient air temperature is, the larger the quantity of heat recovery is, and the lower the inlet water temperature of the heat recovery heat exchanger is, the larger the quantity of heat recovery is. Some studies are focused on the investigation of indoor air quality factors in terms of HVAC systems. In [10], Wang et al. dealt with the influences of an air filter on the performance of a heat recovery ventilator system as well as the application potential reported for this integrated system in five climate zones of China. The results showed that the installation of the air filter (EU F7) was sufficient to decrease the PM_2.5_ concentration and meet the required threshold value of 75 mg/m^3^ specified in the national standard, even though the PM_2.5_ concentration of outdoor air is 820 mg/m^3^. However, the test results indicate that the air flux was reduced by 9.7%–19.9%, and the power of the fan increased by 12.6%–17.5% due to the installation of the air filter. In addition, the case study showed that the integrated system installed in the residential apartment in winter not only improved the IAQ but also saved energy consumption during fresh air heating. Finally, it can be said that improving the building energy efficiency and decreasing the building energy consumption through the monitoring of HVAC and heat recovery systems are becoming increasingly significant due to the global energy crisis and carbon emissions. In [11], Kapalo et al. focused on the ventilation rate and carbon dioxide (CO_2_) concentration. The results showed that the insufficient ventilation intensity in classrooms caused an increase in CO_2_ concentration during exams. The study pointed out that the highest concentration of CO_2_ was recorded during intense physical activity (running on the spot, squats, right and left side lunges, and rotation of the hips). The dependence of indoor air quality on ventilation intensity was also shown in [12]. Due to the high CO_2_ concentration in the indoor environment, the optimal ventilation rate must be calculated. Methods for the determination of the optimal ventilation rate can be found in previous studies [13,14]. A poor IEQ is closely related to sick building syndrome (SBS), which can be a result of exposure to indoor air pollutants, as SBS causes are sometimes unknown. Headache, fatigue, nausea, dizziness, eye, nose and throat irritation, sensation of dry mucous membranes, skin erythema, high frequency of airway infection and coughing, hoarseness, wheezing, and unspecified hypersensitivity are the symptoms of SBS [15]. Zamani et al. [16] investigated the relationship between indoor air quality and the prevalence of SBS in the old and new office buildings in Selangor. They found that an increased ventilation rate can significantly reduce the prevalence of SBS.

In the last decade, several scientific studies have focused on the analysis and simulation of historic buildings with the aim of reducing their energy consumption and increasing their safety. Historic buildings represent a significant cultural property in each country. Therefore, it is important to pay increased attention to these buildings and to ensure their value through renovation and the provision of good indoor environmental quality. Balocco et al. [17] proposed a method to identify the suitability of the indoor environment from historical libraries. Their approach is based on the use of a numerical simulation to resolve the air velocity, moisture, and temperature fields and an assessment of post-processing indexes to evaluate the indoor microclimatic conditions. D’Agostino and Congedo [18] analyzed the microclimatic variations for ventilation scenarios inside the Crypt of Lecce Cathedral through the developed Computational fluid dynamics (CFD) model. The results indicated that natural ventilation can produce different microclimatic conditions and these variations can affect moisture dynamics and artwork conservation. The results also confirmed that CFD modeling can contribute to the provision of better control of the indoor environment, thereby promoting historical building conservation. A study by Balocco and Grazzini [19] investigated the airflow patterns, distribution and velocity, and air temperature distribution inside a historical building in Palermo (Italy) by a transient simulation. A three-dimensional model of a library room with an ancient natural ventilation system was investigated using a CFD tool during the hottest day of summer. The results showed that the CFD modelling approach can provide basic support to the issues involved in the analysis and preventive conservation of ancient ventilation techniques in old buildings. In another study, Kamaruzzanman el al. [20] focused on the IEQ in six historic buildings in Malaysia and presented the possibility of improving the building environment in two ways: Firstly, by providing lessons and feedback for owners or those involved in the environment improvement works. This could lead to enhanced IEQ by addressing the changing needs of occupants. Secondly, it could empower end-users and provide a benchmark and a pool of analyses to show how the building design and its environment management meet the needs of its clients and users. An interesting refurbishment of an historical building, Ca’ S. Orsola, in Treviso with the aim of transforming it into a prestigious residential complex toward nZEB was presented by Mora et al. [21]. The complex library hall renovation of a historical building presented in the research of Rungruengsri [22] showed that the indoor environment can be created to be comfortable even when the area is not provided with an air-conditioning system. This review introduces some examples of the use of structural, energy, and environmental efficiency approaches and the application of heating, ventilation, and air conditioning systems to renovate and protect the historical buildings.

This paper deals with the energy efficiency of ventilation systems in different design alternatives. The alternatives to the operation of mechanical and natural ventilation systems at intervals during the day are presented. The calculation of the airflow is executed in accordance with the laws and standards valid in the EU and Slovakia. Until 1989, the renovation of buildings was not carried out in order to achieve high energy efficiency. There was no legislation, and therefore, the new construction was carried out with very poor thermo-physical parameters. At present, strict legislation in Slovakia places high demand on the construction and reconstruction of buildings. This legislative leap has led to extreme energy savings, especially by insulating the building envelope. This, on the other hand, has caused a significant decrease in the quality of the indoor environment. Neither new construction nor reconstruction are sufficiently prepared to ensure the required indoor environmental quality in relation to energy efficiency. The main contribution of the paper is the analysis of the behavior of a historic building in order to ensure sufficient indoor air quality by installing a modern ventilation system with heat/cooling recovery.

## 2. Materials and Methods

### 2.1. Investigated Building

The main object considered in this study is a historical building situated at a university campus in Košice, Slovakia. The studied building is listed as a historical building from the year 1906 (pre-war building) and is protected by law as a historical monument (Figure 1). Therefore, it is not possible to carry out its full modernization. Masonry bearing and non-bearing vertical constructions are made from burnt bricks with a thickness of 600 mm up to 1000 mm. Structurally, the object is conceived as a combination of a longitudinal two-wing and a transverse three-wing. All bearing walls and partitions as well as the newly implemented walls and ceilings are made from gypsum plastering with interior paint. The floor finishes including the sill are made from polyvinylchloride (PVC). The staircases are made from concrete. The roof structures, roof sheathing, and wood truss are the original structures.

During the years from 2012–2014, the building was renovated. New plaster was applied on the facade and internal walls. To decrease the energy consumption of the building and increase the indoor air quality, the HVAC system was designed.

#### HVAC System Design

The device for the ventilation and air-heating ventilation of classrooms was designed during the renovation. Air supply and air exhaust are ensured by the compact ventilation unit (HVAC unit). The ventilation unit consists of a heat recovery coil, filters, and a supply and exhaust air fan. A water heater with a thermal power of 4.5 kW is designed for air heating. The water heater is placed outside the HVAC unit in an angular duct on the supply fan, and it is equipped with a mixing valve for the mixing of heating water. The HVAC unit’s location is illustrated in Figure 2 and Figure 3. 

The unit works with real air flow with a maximum volume of 1000 m^3^ per hour (277.8 L/s). The air volume flow was determined to meet the needs of fresh air per person according to standard EN 15251:2012 [23]. These figures illustrate the HVAC unit, the air supply duct, and the air exhaust duct in green, blue, and red, respectively. The air supply to room A is secured through the angular galvanized duct conducted in the floor and through the four inlet angular plates with regulation R1. Air exhaust is secured through the exhaust duct in which the exhaust hole is placed, and it is equipped with a drawn-out diffuser with regulation. The duct is placed in the wall approximately 1 m above the floor. The unit is located in the floor pit in the basement with flexible dilatation and vibration from the building structures. The air duct is located in the floor with thermo insulation and hydro insulation. The air supply and air exhaust are conducted through the façade, and then the duct continues in the ground outside the building into the bush. A plastic air duct serves as a rain roof with a sieve. The ventilation system works as an equilibrium pressure system.

Figure 4 depicts a cross-section of the classroom and ducting. The figure shows the air intake from the exterior, starting with the initial air inlet duct, passing through the masonry into the interior, where the duct enters the air handling unit (basic treatment and air filtration). An air heater is placed downstream of the air handling unit. Then, the air is distributed to the room through a duct in the floor terminated by an end supply air distribution element. Figure 5 is a schematic illustration of the air outlet, which is, again, guided in the floor. The drain outlet is located in the wall. The discharge line and the supply line meet in a recovery unit, as shown in Figure 3. 

### 2.2. Methods for Measuring the Temperature in the Ventilation Unit

An investigation of the energy efficiency of the HVAC system was performed during the month of January. Measurements of the outdoor air temperature and air temperature directly in the ventilation unit were conducted during the week at the time of the learning process, from 8:00 to 16:00, using the system of integrated sensors for measuring the temperature. Over the weekend, the measurements were not done, because at that time, the air handling unit was not in operation.

### 2.3. Methods for Measuring the Indoor Environmental Quality Factors

The objective measurement of indoor environmental quality (IEQ) parameters and subjective measurements through questionnaires were performed in a renovated university classroom situated on the basement floor of the investigated building (Figure 6).

The investigation of IEQ was carried out before and during an English lesson in the heating season (March) and lasted 4 h and 40 min. Objective measurements were carried out with the windows and doors closed during the first half-hour, and then the HVAC system was run for an hour and 15 min until the beginning of the lesson. The HVAC system was turned off during the lesson at the request of the teacher, and students could open the windows according their own discretion.

A multifunctional measuring device TESTO 435-4 (Testo Industrial Services, Lenzkirch, Germany) with an IAQ probe was used for the measurement of indoor air temperature, relative humidity, and carbon dioxide (CO_2_) concentration. The measuring range and accuracy was from 0 to 50 °C, ± 0.3 °C for temperature; from 0% to 100% ± 2% for the relative humidity; and from 0 to 10,000 ppm, ± (75 ppm ± 3% of the remaining measurement value) (+1 to +5000 ppm), ± (150 ppm ± 5% of the remaining measurement value) (+5001 to +10000 ppm) for the carbon dioxide concentration. The Vernon–Jokl globe thermometer was used for measurement of the mean radiant temperature. The handheld analyzer of Brüel and Kjaer, type 2250 (Naerum, Denmark) was used to measure the sound pressure level. Concentrations of particulate matter (fractions from 0.5 to 10 µm) were measured using the Handheld 3060 IAQ (Lighthouse Worldwide Solutions, Inc., Landing Parkway Fremont, CA 94538, USA). All measuring devices were placed in the center of the classroom at a height of 1.1 m above the floor (“breathing zone”). The indoor air temperature, relative humidity, CO_2_ concentration, and concentrations of particulate matter were recorded at one-minute intervals. The sound pressure level and mean radiant temperature were recorded at 15 minute intervals. 

Questionnaires were filled by students (12 respondents) at the beginning and end of the lesson. Table 1 shows the scales used for the subjective evaluation. A scale was also used for the overall evaluation of the indoor environmental quality (more acceptable than unacceptable, more unacceptable than acceptable; 0 very tolerable, 1 tolerable, 2 quite tolerable, 3 hardly tolerable, 4 intolerable).

## 3. Results and Discussion

### 3.1. Energy Performance of the HVAC System

The results of the investigated input physical parameters and energy performance of the used HVAC system are presented in Table 2. The determined parameters are presented as the average outdoor temperature (θ_ae_), the average outdoor relative humidity (R_he_), the temperature after heat recovery (θ_hr_), the efficiency of heat recovery (η_r_), the loss by heat recovery ventilation (Ф_r_), the energy consumption for preheating (E_preh_), and the total energy consumption (E_tot_).

As we can see, the temperature varied from 16.0 to 17.2 °C with an average value of 16.4 °C. The average outdoor air temperature ranged from −10.2 to 5 °C with an average value of −1.9 °C. The average outdoor humidity ranged from 63% to 99% with a mean value of 83.8%. The efficiency of heat recuperation was determined to be from 81.4% to 86.6% with an average value of 83.5%.

The results of energy consumption during the operation of the classroom and ventilation unit are illustrated on Figure 7. The energy consumption for each day is divided into four categories:The total energy consumption for the operation of air handling units for one day;The energy consumption for air pre-heating. This is the energy that must be supplied to the pre-heater exchanger. The pre-heater exchanger is turned on when the exterior air temperature is below −9 °C (>−9 °C). This temperature state (exactly −10.2 °C—average exterior air temperature) occurred only on one day, the 4^th^ of January (Monday). On the other days, the pre-heater exchanger was turned off because the exterior air temperature was above −9 °C (<−9 °C).An average exterior air temperature above −8.8 °C (<−8.8 °C) does not cause freezing of condensed water vapor on the inside walls of the air handling unit. In all other cases, the exterior air temperature was above −8.8 °C (<−8.8 °C), except, as already mentioned, the 4^th^ of January.Energy consumption for heating—energy produced from the heat source needed for space heating, which is transferred to the heated space via a duct air heaterElectric energy consumption—energy consumption for fans.

Energy consumption for the reheating of air ranges from 5.6 to 7.8 kWh/day, and the average value is 7.1 kWh/day. Energy consumption for the fan operation is 2.3 kWh/day. Energy consumption for the preheating of air is 8 kWh/day. Finally, the total energy consumption for the operation of air handling units varies between 7.9 and 18.4 kWh/day with average value of 9.8 kWh/day.

### 3.2. Indoor Environmental Quality in the Investigated Building

Table 3 presents the IAQ parameters measured in the classroom before the beginning of the lesson and during the lesson. The mean indoor air temperature during the lesson was about 0.4% lower during the lesson than before the beginning of the lesson. The relative humidity during the lesson was about 8.1% higher during the lesson than before the beginning of the lesson. The difference between the CO_2_ concentrations was 48.7%. The mean CO_2_ concentration before the beginning of the lesson was 538.7 ppm and during the lesson was 1050.5 ppm. The PM concentration and sound pressure level were also higher during the lesson than before the beginning of the lesson. 

The mean measured indoor air temperature was 24.8 °C; therefore, students felt slight heat at the beginning and also at the end of the lesson. According to the Decree of the Ministry of Health Slovak Republic No. 259/2008 Coll., the relative humidity should be in the range of 30%–70%. In this case, the mean relative humidity was 31%. Students evaluated humidity as neutral, but the results were slightly worse at the end of lesson. Air draught was felt by students as slight at the beginning and also at the end of the lesson. The mean concentration of CO_2_ was 1050.5 ppm but maximum the concentration reached a value of 1641 ppm, which is higher than the recommended value of 1000 ppm by about 39% according to Pettenkofer [24]. In a study by Gao et al., the mean CO_2_ concentration in the classroom ventilated primarily by mechanical ventilation was 954 ppm [25]. This is about 9.2% lower compared with our concentration of 1050.5 ppm. Much higher CO_2_ concentrations were measured in the study of Klavina et al., which were focused on indoor air quality in renovated schools in Latvia. The mean CO_2_ level at the beginning of the lesson was 1156 ppm, and at the end of the lesson, it was 2579 ppm. The mean air temperature was 19.2 °C and 22.6 °C at the beginning and at the end of the lesson, respectively. The most commonly occurring symptom among students was also fatigue [26]. In study of Batterman, CO_2_ levels reached 2100 ppm during the school day [27]. In study of Pinto [28] threshold of CO_2_ protection exceeded (1625 ppm for a mean of 8 h). Another factor, odor, was also felt by students as slight at the beginning as well as at the end of the lesson. The mean PM_10_ concentration was 27.0 μg.m^−3^. Lower concentrations of PM_10_ were measured by Fischer et al. in a mechanically ventilated classroom. The average PM_10_ concentration was 14 μg.m^−3^. A study found that the PM_10_ concentration was the largest in occupied, ventilated classrooms, and this is controlled by human presence and indoor activities as well as the particle number concentration depending on the outdoor conditions [29]. On the other hand, higher concentrations of PM_2.5_ (23 μg.m^−3^) and PM_10_ (105 μg.m^−3^) were found in 92 naturally ventilated classrooms in the northern part of the city of Munich and in a neighboring rural district. This study stated that secondary inorganic aerosols were the dominant water-soluble ions of indoor and outdoor PM. The study also noted that the smaller the particle, the higher the percentage of secondary inorganic aerosols. Except for PM_10_ in the school dormitory, strong correlations were found between indoor and outdoor PM [30]. Rufo J. C., et al. [31] investigated how IAQ changed in primary schools after applying indoor air quality recommendations and explored how these changes influenced allergic sensitization in children. The recommendations were based on the guidelines for healthy environments within European schools. For instance, if PM concentrations in a classroom were very high and the blackboard was identified as a main emission source, the recommendations would suggest more frequent blackboard cleaning with a wet cloth to prevent particle suspension. However, there was no supervision to verify if the recommendations were correctly adopted. Indoor PM_2.5_ and PM_10_ concentrations were approximately 40% lower in the follow-up measurements. In our previous study, the mean indoor air temperature, relative humidity, and concentrations of PM_2.5_ and PM_10_ were 24.86; 45.83%; 12.40 μg.m^−3^; and 108.90 μg.m^−3^, respectively, in the university classroom with natural ventilation. Linear dependence was found between concentrations of PM_2.5_, PM_5_, and temperature. The correlation was almost perfect between the concentration of PM_5_ and the temperature [30]. The permissible noise level for classrooms is 40 dB [32]. The calculated mean value of the equivalent sound pressure level was about 35.1% (61.6 dB) higher than the permissible level according to the Decree of the Ministry of Health of the Slovak Republic No. 549/2007 Coll. [33]. Students also evaluated noise as acceptable at the beginning of the lesson and as slight at the end of the lesson. Due to the noise levels of HVAC, the teachers often turn off the HVAC, which leads to inadequate ventilation, poor thermal conditioning, as well as to poor indoor air quality. Spears et al. [34] observed a similar tendency in the behavior of teachers in the studied school building. Lighting was evaluated by students as acceptable. 

The course of measured values of the indoor air temperature, relative humidity, concentration of CO_2_, and concentration of PM for each fraction are shown in Figure 8, Figure 9 and Figure 10). As can be seen, the dynamic changes in the measured indoor environmental parameters were directly affected by the HVAC system and open windows at the end of the lesson. The PM concentrations fluctuated throughout the measurement period, but at the end of lesson, they were higher due to the higher movement of students in the classroom and also due to the open windows. 

Pearson’s correlation analysis was performed to investigate the statistical relationship between measured variables. A correlation ratio R below 0.1 is trivial, from 0.1 to 0.3 is small, from 0.3 to 0.5 is medium, and above 0.5 is high, according to Cohen [35]. A correlation of 0.7 to 0.9 is often cited as a very large and 0.9–1 as almost perfect. An analysis was performed with the STATISTICA software (Dell Inc., Round Rock, Texas, USA) which revealed a very large positive correlation between the relative humidity and the concentration of CO_2_ as well as between the concentration of PM_5_ and the concentration of CO_2_.

A high positive correlation was found between the concentrations of CO_2_ and PM_1_, PM_2.5_ and PM_10_ as well as between the PM_5_ concentration and the relative humidity. A high negative correlation was found between the temperature and the PM_2.5_ concentration. In [36], Lazović et al. detected high correlations between the CO_2_ concentration and PM_2.5_ and PM_10_ in two schools in the non-heating period. They also revealed a correlation between the CO_2_ concentration and relative humidity and indoor air temperature in their next study [37]. A significant negative correlation between humidity and the PM_2.5_ concentration in winter and a significant positive correlation in summer were found in research performed in 64 schools in Germany [38]. A significant positive correlation between the indoor temperature and PM_10_ together with a correlation of CO_2_ and PM_10_ was also found in another study by Alshitawi et al. [39]. Table 4 shows the correlation matrix. Relationships between environmental parameters are depicted in Figure 11.

As Figure 11a shows, the average value of PM_1_ had a linear relationship with CO_2_ (R = 0.51), which is, according to Cohen [35], defined as a high correlation. The same correlation with the value R = 0.51 was found between PM_2.5_ and the temperature (Figure 11g). A very similar correlation (R = 0.56) was found between PM_2.5_ and CO_2_ (Figure 11b) and (R = 0.58) between PM_10_ and CO_2_ (Figure 11d). A very large correlation (R = 0.72) was found between PM_5_ and CO_2_ (Figure 11c). The same large correlation R = 0.72) was found between PM_5_ and the relative humidity (Figure 11e). The largest correlation (R = 0.86) in our study, but still defined as a large correlation according to Cohen [33], was found between the relative humidity and the CO_2_ concentration (Figure 11f).

In our previous study [40] on IEQ in Houses of Macedonia we found correlations between smoke and TVOC; smoke and PM_2.5_ as well as hidden dependence between renovation and smoke.

Twenty-five percent of respondents were men with a mean height of 178.3 m and a mean weight of 92.7 kg, and 75% were women with a mean height of 165.7 m and a mean weight of 60.4 kg. Only one correspondent was a smoker. Thirty-three percent of respondents had an allergy. As can be seen in Figure 12, only a few respondents complained about SBS symptoms. The limiting factor of this study was the small number of questionnaires completed. The most commonly occurring symptoms were fatigue and a feeling of a heavy head. 

The results of the sensorial evaluation at the beginning and at the end of lesson are illustrated in Figure 13. It can be seen that there were obvious minimal differences between the perception of the indoor environmental quality at the beginning and at the end of lesson. The most significant difference was in the perception of the noise level, which was perceived to be better at the end of the lesson. 

Turunen et al. [41] presented a study of the school- or grade-level prevalence of symptoms in relation to IEQ according to a health questionnaire. Questionnaires were sent to all 6th grade students in a stratified random sample of 355 elementary schools in Finland. The most common weekly symptoms in the spring term were fatigue (7.7%), stuffy nose (7.3%), and headache (5.5%). In our previous study [42] on the IEQ of classrooms and occupants’ comfort in a special education school, we stated that the most common SBS symptoms in students (indicated by more than 40% of students) were observed to be fatigue, feeling heavy-headed, headache, difficulty with concentration, eye irritation, and nasal symptoms as well as a sore throat.

Sakelleris et al. [43] presented the relations between perceived indoor environment and occupants’ comfort depend on socio-cultural context, as well as personal and building characteristics. Study state that understanding and evaluating occupants and building characteristics may help in providing healthier and more comfortable buildings.

Our future work will be aimed at the investigation of a significant set of low energy buildings to determine the indoor environmental quality by objective monitoring as well as the sensational perception of occupants. The standard operation of indoor spaces is characterized by the fluctuations and balance of indoor environmental loading, which have been taken into account, as well as energy performance (energy savings, energy losses). From this reason, it is necessary to apply temperature, humidity, and CO_2_ concentration sensors in rooms with people staying in order to ensure an acceptable indoor air quality. There is a real assumption that application of the proper measuring and control technology can reduce the minimum value of air exchange because the ventilation technology will also be able to ensure the required air quality is present by means of the reduced air exchange. A full-scale computer (FSC) model of the investigated building will be created in the future to allow a comprehensive energy consumption analysis. The ambient temperature, air speed, thermo-physical properties of the components, the spectral characteristics of the glazing system and the solar gains as well as the air exchange rate and energy demand for HVAC systems are the important variables for FSC modelling.

## 4. Conclusions

A historical building was studied in terms of its energy performance and indoor environmental quality. The objective of this case study was to analyze the efficiency of a uniquely designed HVAC system, determine the level of physical factors in terms of their impacts on the health of building users, and find out the relationship between environmental parameters and compare measured values via subjective evaluation through questionnaires. The results show that the mean relative humidity (31%) was in the permissible range (30%–70%); the mean CO_2_ concentration (1050.5 ppm) was above the recommended value of 1000 ppm; and the mean PM_10_ concentration (43.5 µg/m^3^) is under the limit value of 50 µg/m^3^. The correlation analysis revealed a relationship between measured factors. A very large positive correlation was found between the relative humidity and the concentration of CO_2_ as well as between the concentrations of PM_5_ and CO_2_. The most commonly occurring SBS symptoms are fatigue and the feeling of a heavy head.

In conclusion, it should be emphasized that historic buildings are part of each country’s national and cultural identity. The main prerequisite for the protection of historic buildings is knowledge and understanding of their value. The improvement of the technical characteristics and functionality of historic buildings and their overall revitalization can extend their lifetime and lead to social, economic, and environmental benefits.

## Figures and Tables

**Figure 1 ijerph-16-04133-f001:**
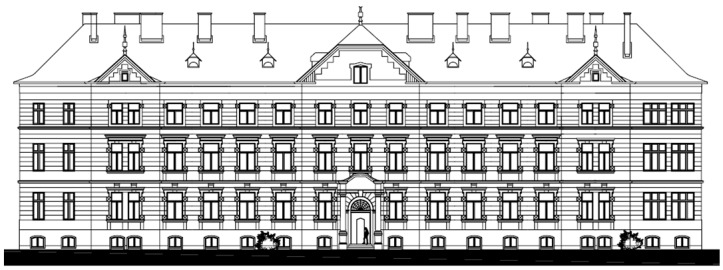
View of the building.

**Figure 2 ijerph-16-04133-f002:**
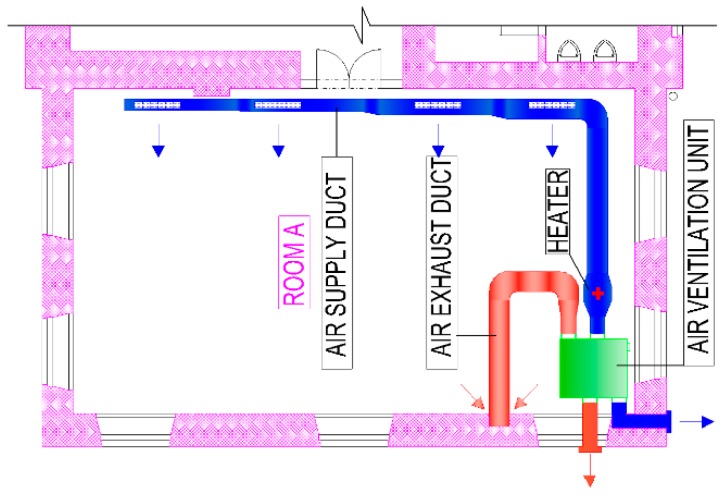
Floor plan of the investigated classroom with the designed heating, ventilation, and air conditioning (HVAC) system marked.

**Figure 3 ijerph-16-04133-f003:**
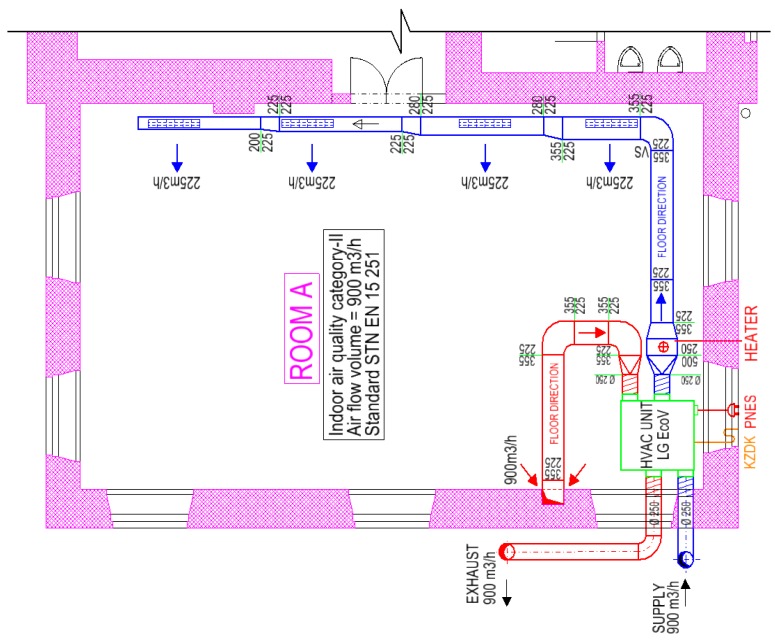
Floor plan of the investigated classroom with quantities of supplied and exhausted air.

**Figure 4 ijerph-16-04133-f004:**
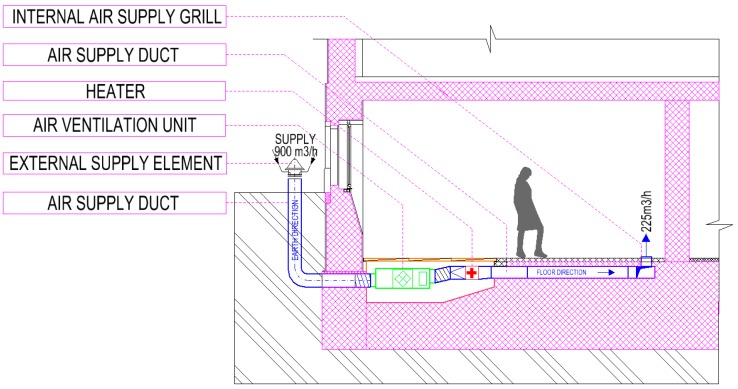
Cross-section of the investigated room with the designed ducts of the HVAC system.

**Figure 5 ijerph-16-04133-f005:**
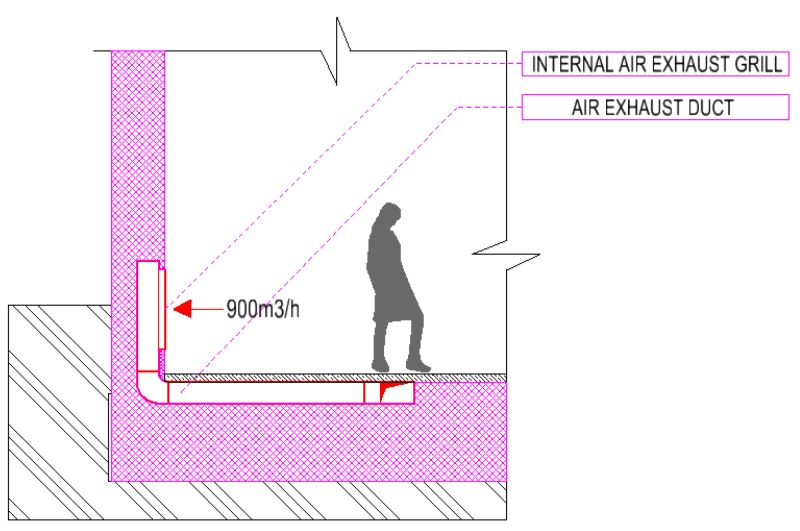
Cross-section of the investigated room with specified quantities of supplied air.

**Figure 6 ijerph-16-04133-f006:**
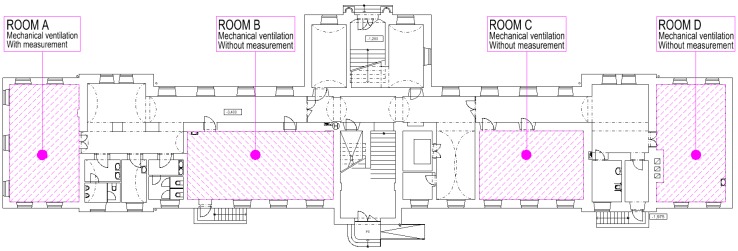
Plan of the basement.

**Figure 7 ijerph-16-04133-f007:**
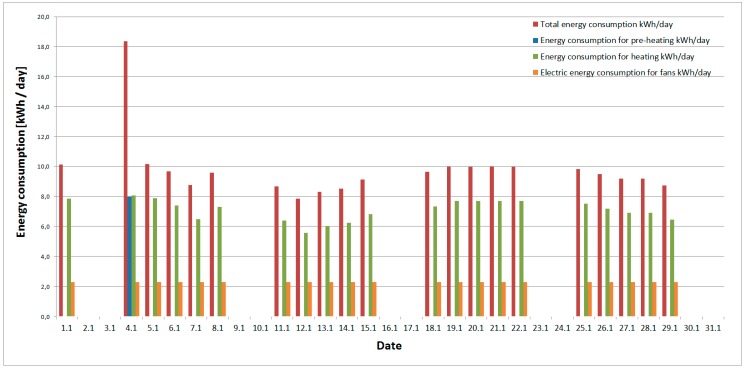
Energy consumption.

**Figure 8 ijerph-16-04133-f008:**
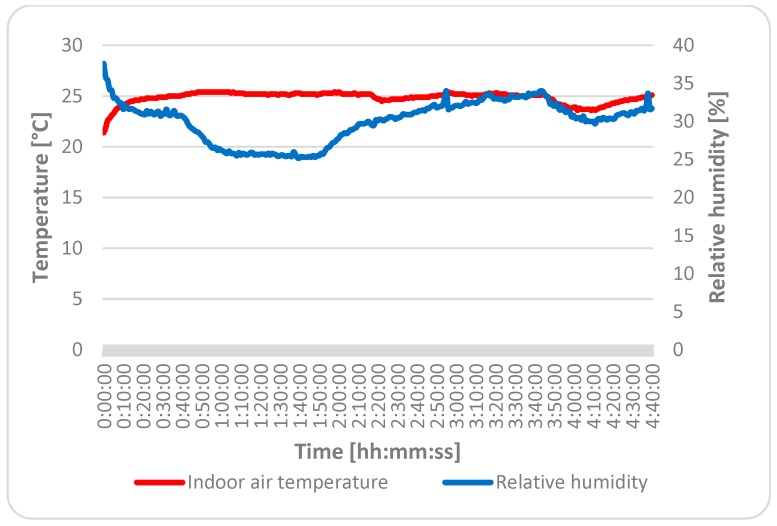
Course of measured values of indoor air temperature and relative humidity.

**Figure 9 ijerph-16-04133-f009:**
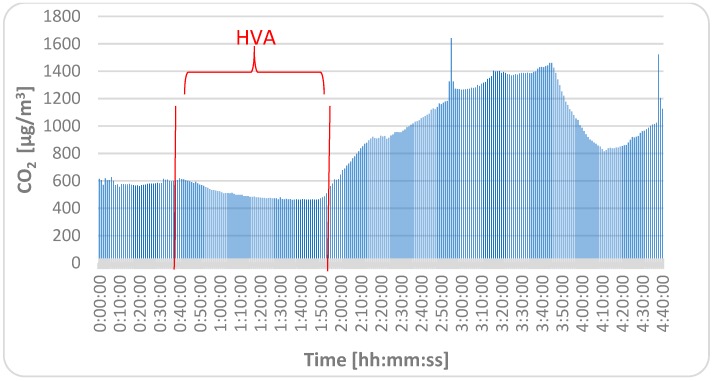
Course of measured concentrations of CO_2_.

**Figure 10 ijerph-16-04133-f010:**
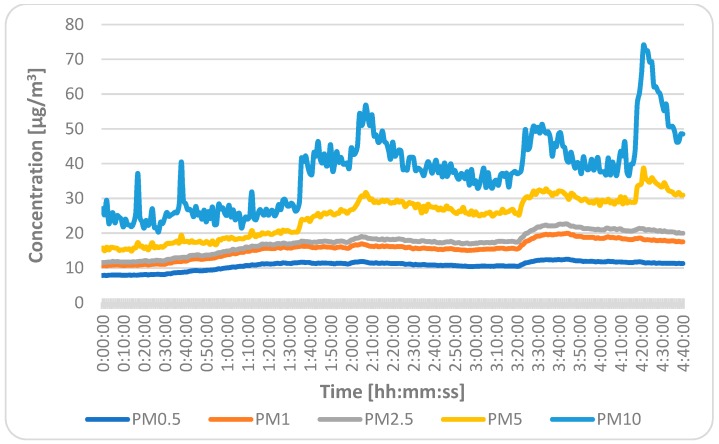
Course of dynamic changes of particulate matter concentrations.

**Figure 11 ijerph-16-04133-f011:**
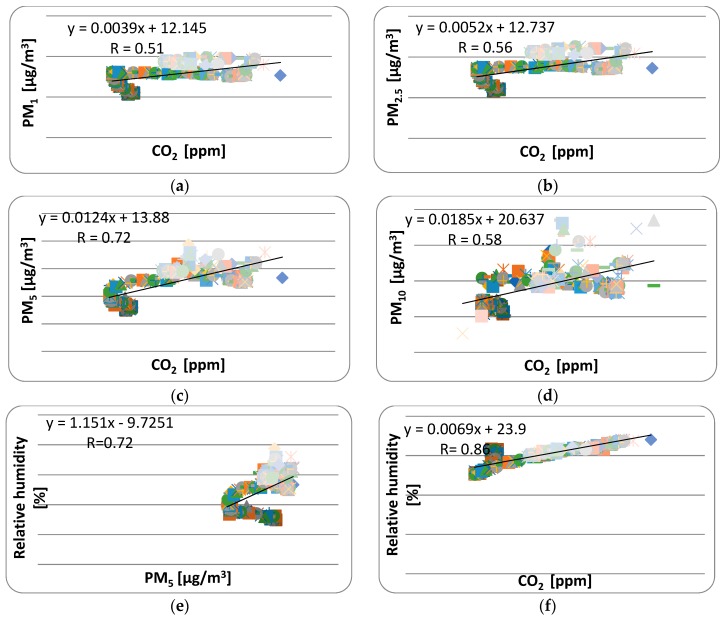
Correlations between environmental parameters. (**a**) Correlation between CO_2_ and PM_1_; (**b**) Correlation between CO_2_ and PM_2.5_; (**c**) Correlation between CO_2_ and PM_5_; (**d**) Correlation between CO_2_ and PM_10_; (**e**) Correlation between PM_5_ and RH; (**f**) Correlation between CO_2_ and RH; (**g**) Correlation between temperature and PM_2.5_.

**Figure 12 ijerph-16-04133-f012:**
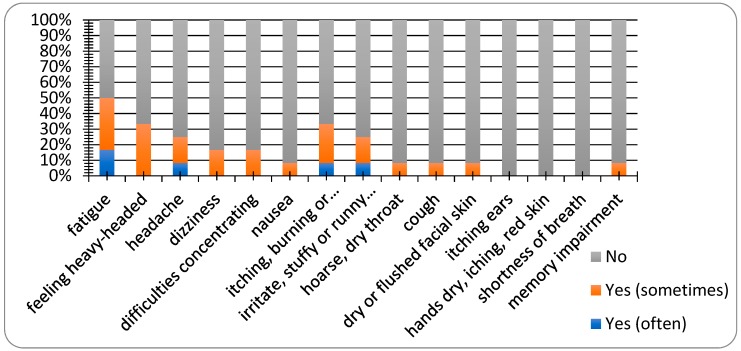
Evaluation of sick building syndrome (SBS) symptoms.

**Figure 13 ijerph-16-04133-f013:**
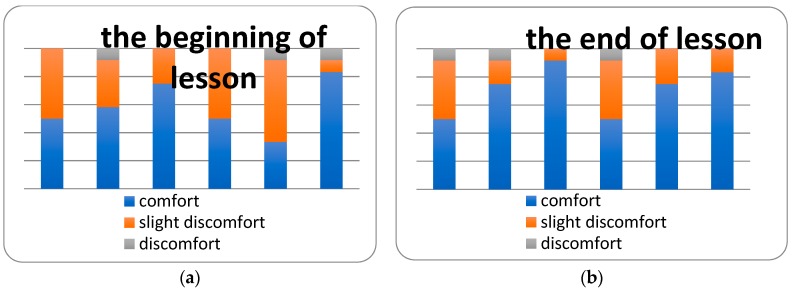
Sensory evaluation. (**a**) Evaluation at the beginning of the lesson; (**b**) Evaluation at the end of the lesson.

**Table 1 ijerph-16-04133-t001:** Scales used for the subjective evaluation.

	Perception	Sensational Evaluation
Humidity	2 humidity 1 slight humidity 0 neutral –1 slightly dry –2 dry	0 comfort 1 slight discomfort 2 discomfort 3 strong discomfort
Indoor air temperature	3 hot 2 warm 1 slightly warm 0 neutral –1 slightly cool –2 cool –3 cold	0 comfort 1 slight discomfort 2 discomfort 3 strong discomfort
Air draught	0 no air draught 1 slight air draught 2 mild air draught 3 strong air draught 4 very strong air draught 5 sublime air draught	0 comfort 1 slight discomfort 2 discomfort 3 strong discomfort
Indoor air quality	0 no odor 1 weak odor 2 moderate odor 3 strong odor 4 very strong odor 5 overpowering odor	0 comfort 1 slight discomfort 2 discomfort 3 strong discomfort
Noise	1 no noise 2 low noise 3 acceptable noise 4 strong noise 5 intolerable noise	0 comfort 1 slight discomfort 2 discomfort 3 strong discomfort
Lighting	2 very high 1 high 0 acceptable –1 low –2 very low	0 comfort 1 slight discomfort 2 discomfort 3 strong discomfort

**Table 2 ijerph-16-04133-t002:** Input physical parameters for energy performance calculation.

Date	θ_ae_ [°C]	R_he_ [%]	θ_hr_ [°C]	η_r_ [%]	(Φ_r_) [W]	E_preh_ [kWh/day]	E_tot_ [kWh/day]
1.1	−6.2	72	16.1	85.0	980.0	7.8	10.1
4.1	−10.2	77	16.0	86.6	1009.0	8.1	10.4
5.1	−7.1	83	16.0	85.4	984.2	7.9	10.2
6.1	−2.9	84	16.3	83.8	923.9	7.4	9.7
7.1	1.6	74	16.7	82.3	811.0	6.5	8.8
8.1	−2.3	90	16.3	83.6	912.6	7.3	9.6
11.1	2.0	99	16.8	82.2	799.3	6.4	8.7
12.1	5.0	86	17.2	81.4	696.2	5.6	7.9
13.1	3.3	82	17.0	81.9	751.6	6.0	8.3
14.1	2.5	81	16.9	82.1	779.5	6.2	8.5
15.1	0,2	94	16.6	82.7	853.5	6.8	9.1
18.1	−2.6	63	16.3	83.7	918.3	7.3	9.6
19.1	−5.1	71	16.1	84.6	963.5	7.7	10.0
20.1	−4.9	79	16.1	84.5	961.2	7.7	10.0
21.1	−5.1	72	16.1	84.6	963.5	7.7	10.0
22.1	−5.1	71	16.1	84.6	961.4	7.7	10.0
25.1	−3.6	94	16.2	84.0	941.3	7.5	9.8
26.1	−1.6	96	16.4	83.3	899.2	7.2	9.5
27.1	−0.2	99	16.5	82.8	864.2	6.9	9.2
28.1	0.0	98	16.5	82.7	862.1	6.9	9.2
29.1	1.7	95	16.8	82.3	805.6	6.4	8.7

**Table 3 ijerph-16-04133-t003:** Measured Indoor Environmental Parameters.

Parameter	Before the Beginning the Lesson				During the Lesson				Limit/Recommended Value
	Mean	Min.	Max.	S.D.	Mean	Min.	Max.	S.D.	
θai [°C]	24.9	21.4	25.4	0.7	24.8	23.6	25.4	0.5	-
RH [%]	28.5	25.2	37.6	3.0	31.0	25.3	34	2.0	30–70
CO_2_ [ppm]	538.7	461.0	627	53.9	1050.5	463	1641	271.1	1000
PM_0.5_ [µg/m^3^]	9.6	7.8	11.7	1.4	11.4	10.4	12.5	0.6	-
PM_1_ [µg/m^3^]	13.2	10.6	16.3	2.0	17.0	15.0	20.0	1.8	-
PM_2.5_ [µg/m^3^]	14.3	11.6	17.7	2.1	19.3	16.9	22.7	1.8	-
PM_5_ [µg/m^3^]	18.3	14.9	25.7	2.5	28.9	24.7	38.7	2.7	-
PM_10_ [µg/m^3^]	27.0	20.3	46.4	5.2	43.5	32.9	74	8.1	50
LA_eq._ [dB]	59.9	58.1	62.1	1.7	61.6	56.0	68.6	3.7	40

**Table 4 ijerph-16-04133-t004:** Correlation matrix (*p* < 0.05).

	CO_2_	θ_ai_	RH	PM_0.5_	PM_1_	PM_2.5_	PM_5_	PM_10_
CO_2_	1.00	−0.14	0.86	0.42	0.51	0.56	0.72	0.58
θ_ai_	−0.14	1.00	−0.30	−0.34	−0.49	−0.51	−0.42	−0.26
RH	0.86	−0.30	1.00	0.09	0.25	0.33	0.53	0.46
PM0.5	0.42	−0.34	0.09	1.00	0.96	0.94	0.83	0.67
PM1	0.51	−0.49	0.25	0.96	1.00	1.00	0.89	0.72
PM2.5	0.56	−0.51	0.33	0.94	1.00	1.00	0.92	0.76
PM5	0.72	−0.42	0.53	0.83	0.89	0.92	1.00	0.91
PM10	0.58	−0.26	0.46	0.67	0.72	0.76	0.91	1.00

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
