# Peer review of "Investigation of a Ventilation System for Energy Efficiency and Indoor Environmental Quality in a Renovated Historical Building: A Case Study"

_ijerph, 2019, doi:10.3390/ijerph16214133_

Round 1

Reviewer 1 Report

This paper needs thorough screening in all its sections. Below are few of the problems seen in the paper.

Abstract: 

line 14. pls replace 'the paper' with 'this paper'

line 17. put IEQ in bracket at first mention of indoor environmental quality

line 19. Pettenkofer is not mentioned in the references. There are international standards that can be quoted such as ASHRAE standard 62.1: 2016, ventilation for acceptable indoor air quality.

Introduction

line 28 to 36. put appropriate ref.

line 36. the sentence ‘among the diff. building type….’ should have a ref.

line 43. according to study 6 should be re-written to convey meaning. Kindly write as e.g. ‘according to a study by X [7],… and not according to [7].

In all, there is no cohesion in the introduction, paragraphs are not properly used to separate important points and some stated points/facts are not referenced.

SBS was not mentioned throughout the introduction section and the keywords are not properly defined or introduced.

The use of English language is below average, the entire introduction should be re-written.

Materials and methods.

The location of the building under investigation was not stated.

when was the building constructed and give the major chrematistics that makes the building historical apart from its protection by law?

line 97. give full meaning of PVC at first mention.

Explain the relationship between the historical building and classrooms.

line 112. please give ref. to the standard.

line 154. which month is the heating season?

line 158. how did you know that the windows had no significant impact on airflow?

The manufacturing company and country for measuring device(s) not stated.

Can you give figure(s) to show some examples of questions asked in the questionnaire?

was the questionnaire validated?

Results and discussion

The period of examining the building was not stated.

The statistical analysis method and software used for the results was not stated and explained at the beginning of the result section or in the method section.

There is inadequate ref and discussion. Results and discussion can be separated for better writing and understanding.

The writers are advised to read more published papers in this field to be able to write effectively. The writers may also employ a native English language speaker. This paper as it is cannot be published.

Author Response

We would like to thank for the valuable advice and suggestions.

This paper needs thorough screening in all its sections. Below are few of the problems seen in the paper.

Abstract: 

line 14. pls replace 'the paper' with 'this paper'

Response: corrected

line 17. put IEQ in bracket at first mention of indoor environmental quality

Response: corrected

line 19. Pettenkofer is not mentioned in the references. There are international standards that can be quoted such as ASHRAE standard 62.1: 2016, ventilation for acceptable indoor air quality.

Response: Reference is added.

Introduction

line 28 to 36. put appropriate ref.

Response: Sentences are rephrased.

line 36. the sentence ‘among the diff. building type….’ should have a ref.

Response: Sentences are rephrased.

line 43. according to study 6 should be re-written to convey meaning. Kindly write as e.g. ‘according to a study by X [7],… and not according to [7].

Response: Citations are checked.

In all, there is no cohesion in the introduction, paragraphs are not properly used to separate important points and some stated points/facts are not referenced.

SBS was not mentioned throughout the introduction section and the keywords are not properly defined or introduced.

The use of English language is below average, the entire introduction should be re-written.

Response: Introduction is corrected. The manuscript will be sent for English editing to professional English editing service.

Materials and methods.

The location of the building under investigation was not stated.

Response: Location of the building is added (line 164-165).

when was the building constructed and give the major chrematistics that makes the building historical apart from its protection by law?

Response: Building was built in 1906 – line 165.

line 97. give full meaning of PVC at first mention.

Response: The full meaning of PVC is added (line 171).

Explain the relationship between the historical building and classrooms.

Response: Classrooms in which the research was performed are placed in the historical building.

line 112. please give ref. to the standard.

Response: Reference is added (187).

line 154. which month is the heating season?

Response: In Slovakia, the heating season begins when the outside air temperature drops below 13°C for two consecutive days. The month is added - line 230.

line 158. how did you know that the windows had no significant impact on airflow?

Response: We deleted the sentence about this statement.

The manufacturing company and country for measuring device(s) not stated.

Response: Information are added - lines 237, 244, 246.

Can you give figure(s) to show some examples of questions asked in the questionnaire?

was the questionnaire validated?

Response: I put here the screenshot (Fig. 1) of the questionnaire. The questionnaires were validated by experts of the specialized institute for the questionnaire survey.

Figure 1. Questionnaire for investigation of IEQ perception by students (In Slovak)

Results and discussion

The period of examining the building was not stated.

Response: The period of examining the building is from January to March – lines 217, 230.

The statistical analysis method and software used for the results was not stated and explained at the beginning of the result section or in the method section.

Response: Information about statistical analysis are added – lines 353 - 356.

There is inadequate ref and discussion. Results and discussion can be separated for better writing and understanding.

Response: Thank you for your advice. We tried to separate results and discussion. But the text lost its fluidity. Information flow disrupted.

The writers are advised to read more published papers in this field to be able to write effectively. The writers may also employ a native English language speaker. This paper as it is cannot be published.

Response - Thank you very much for your recommendations. The manuscript will be sent to professional English editing service.

Reviewer 2 Report

A case study is presented to calculate the energy consumption of an HVAC system and it's impact on the internal air quality. The reviewer has the following suggestion to improve:

The title needs to be changed as it is confusing for the reader. The reviewer was not able to find out a significant comparison of mechanical +natural ventilation (mixed-mode) systems in the study. The reference was made to an earlier study [18] for the particulate values but it is not sufficient to highlight in the title as natural ventilation modeling.  The reference to energy modeling is also not significant as most of the study has conducted by physically measuring the values with handheld devices. If some computer modeling was done, it is needed to be explained in the materials and methods.  Authors should include a comprehensive literature review after ty introduction to explain if similar studies have been done in the region and how this study is adding to the existing knowledge.  The authors need to present physical characteristic of the building itself (materials, envelop details etc.). Please provide a table with the range of values for IEQ standards used for this study. Also, is there a new version of the standard available as the reference has been made to 2007 version (EN 15251:2007). What are the limitations of the study? For the replicability purposes, it is important to understand if there was a limited data or some aspects were normalized due to other reasons.  The study was presented for a room in the educational building. What about the impact of air leakages and convection from the adjacent rooms? Are authors going to create a full-scale computer model of the building in future to have a comprehensive energy consumption analysis (different types of rooms, different purpose of the room, diverse occupants etc.)?

Author Response

Comments and Suggestions for Authors

A case study is presented to calculate the energy consumption of an HVAC system and it's impact on the internal air quality. The reviewer has the following suggestion to improve:

The title needs to be changed as it is confusing for the reader.

The reviewer was not able to find out a significant comparison of mechanical +natural ventilation (mixed-mode) systems in the study. The reference was made to an earlier study [18] for the particulate values but it is not sufficient to highlight in the title as natural ventilation modeling. The reference to energy modeling is also not significant as most of the study has conducted by physically measuring the values with handheld devices. If some computer modeling was done, it is needed to be explained in the materials and methods. 

We would like to thank for the valuable advice and suggestions.

Response: Title is changed.

Authors should include a comprehensive literature review after ty introduction to explain if similar studies have been done in the region and how this study is adding to the existing knowledge. 

There is a lack of studies of energy efficiency of HVAC systems in historical buildings in our region. Renovation and application of special ventilation systems in historical buildings is unique in Slovakia. That's why we chose this building for research.

We added more studies about historic buildings and their renovation in terms of energy efficiency improving and application of heating, ventilation and air conditioning systems,…

The authors need to present physical characteristic of the building itself (materials, envelop details etc.).

Response: Characteristic of the building is added – lines 167 – 172.

Please provide a table with the range of values for IEQ standards used for this study.

We added column to Table 3 with limit values /recommended values according to Slovak legislative. In the text below the Table 3 the legislative is presented.

Also, is there a new version of the standard available as the reference has been made to 2007 version (EN 15251:2007).

Thank you for your note. The year of the standard is corrected.

What are the limitations of the study? For the replicability purposes, it is important to understand if there was a limited data or some aspects were normalized due to other reasons. 

Response: Thermo-physical parameters of building structures are considered as standard values according Slovak standards valid for the given period.

Factors of indoor and outdoor environment were determined by measurement using analysers and instruments.

Measuring of indoor environmental parameters was performed during normal operation of classrooms.

The study was presented for a room in the educational building. What about the impact of air leakages and convection from the adjacent rooms? Are authors going to create a full-scale computer model of the building in future to have a comprehensive energy consumption analysis (different types of rooms, different purpose of the room, diverse occupants etc.)?

Yes, the authors aim to model the building for a comprehensive analysis of the energy performance of the building. We added the sentence about it – lines 415 416.

The manuscript will be sent to professional English editing service.

Reviewer 3 Report

Title: Energy performance modelling of mechanical and natural ventilation and indoor environmental quality in renovated historical building: a case study

Journal: The International Journal of Environmental Research and Public Health

-Reviewer

This paper takes practice to highlight the importance of environmental protection regarding the reduction of energy consumption while keeping the living standard. The paper should be revised (minor revision). The comments are as followed:

There is an insufficient literature review and insufficient detail in the methods section. There is no the introductory paragraph to some sections. Please write opening paragraphs for them. Please change colors in plan view of basement (Figure 6).

Author Response

This paper takes practice to highlight the importance of environmental protection regarding the reduction of energy consumption while keeping the living standard. The paper should be revised (minor revision). The comments are as followed:

There is an insufficient literature review and insufficient detail in the methods section. There is no the introductory paragraph to some sections. Please write opening paragraphs for them.

We would like to thank for the valuable advice and suggestions. Introduction and methods are revised.

Please change colors in plan view of basement (Figure 6).

Response: Colors in figure 6 are changed. 

The manuscript will be sent to professional English editing service.

Round 2

Reviewer 1 Report

Very little was done to improve the manuscript. The use of English language is extremely low for a scientific article. Results presented without adequate discussion.

Line 33 to 35 – pls put a ref.

Line 37 – pls write ‘ most of the energy’

Line 39 – pls remove ‘(2016)’. Also remove from other places too since you are using numbers as ref.

Line 55 – pls remove one ‘E’ from experimental.

Line 77 – define CO2 at first mention.

In the study [] is too much in the write up and it affects the writing flow. Kindly read published papers to fix this.

Line 78 – replace ‘increase’ with concentration and ‘harder’ with ‘intense’

Line 81 – pls replace indoor air quality with the abbreviation (IAQ) you already defined. Do same for others.

Line 83 to 84 – pls write ‘which can be a result of’ as SBS cause(s) are sometimes unknown

Line 87 – remove ‘WHO 1988’, else you are quoting twice

Line 89 and 90 – pls rewrite the sentence

Line 109 and 110 – pls rephrase the sentence

Line 116 – write ‘with the aim’

Line 123 – write ‘this paper’

Line 126 to 128 – pls rephrase

Line 130 – pls replace installation with installing

Line 133 – start with ‘The studied building is listed as a historical building ……’

Line 162 – write ‘dusts are’

Line 163 – ‘Air supply and exhaust are’

Line 165 – you already explained this in the previous sentence

Kindly explain wat you mean by conservationist or remove the sentence about this

Line 175 – pls rephrase the sentence

Label fig 4 and 5 for clarity

Line 185 – write ‘the investigation of HVAC system for energy…….’

Line 186 – replace the comma with ‘and’

Line 189 – write ‘were not done’

Line 192 – write ‘through’ and not throughout

Fig 6. Should be made brighter

Fig 6. Was measurement only done in room A?

Line 198. Write ‘with the windows and doors closed’

Line 200. ‘up to the beginning’

Line 213. Kindly put the correct sign for quotation for ‘breathing zone’

Line 217. Pls remove ‘only’

Line 218. Just write Table 1 shows the scales of the subjective evaluation

Line 226. Write ‘the determined parameters are presented as’

Line 228. What is ‘loss by heat recovery ventilation’?

Line 233. Is it the measured temp. or average temp?

This is not clear from the write up and the table. Can you explain why you have average RH and average outdoor temp and the others are not average?

Fig 7. Correct the spelling for ‘consumption’.

Line 241. Pls rewrite the sentence

Line 245. Pls rewrite ‘which is needed to prevent of water vapour condense’

The four categories mentioned in the fig (fig 7) and those mentioned from line 241 to 247 differs. E,g., electric energy consumption for fans.

Line 249. Pls rewrite the sentence ’The average outdoor temp. in borders of…’ do same for the sentence that starts with ‘The reason was that’

Line 255. Pls remove ‘is’

Line 258. Pls rewrite the sentence

Line 262. Pls remove ‘more significant’ except you want to explain how it was more significant.

Line 263 and 264. Pls remove 28% or write the % for PM concs. Too

Line 273. How did you link air quality with co2 conc? How did you know that odour was caused by VOCs?

Line 283. Pls write PM10 well and do same at other places too.

You did not discuss why PM was low in some studies and why it was high in others. What can be done to make it stay within recommendation?

Line 295 put ref. for permissible noise level.

Line 297. Pls put the ref. in the reference list.

Line 298. You evaluated lighting and not lightning.

Line 298 to 302. Pls rearrange the sentences. Put the sentence that starts with ‘Due’ first, ‘Spears et al.’ second and then ‘lighting’

Line 308. Pls write ‘movement’

Line 325. Pls rewrite

Line 317 to 345. Pls discuss what the correlations depicts/show

Line 347. Pls rewrite ‘only one correspondent was smoker’

Line 348. Pls remove ‘The’

Very weak discussion of results. Limiting factors were also not discussed.

Line 360. Pls remove ‘investigate’ and put ‘a’

Line 362. You did not study chemical factors.

Line 366. Pls remove ‘according to Pettenkofer’

Line 367 and 368. Pls remove ‘performed using statistica software’

Line 369 to 379 should be in the discussion section

Author Response

Comments and Suggestions for Authors

Very little was done to improve the manuscript. The use of English language is extremely low for a scientific article. Results presented without adequate discussion.

Thank you for your valuable comments and suggestions as well as English corrections. Manuscript will be given for English editing to professional service.  We tried to improve our manuscript adding more discussion of our results.

Line 33 to 35 – pls put a ref.

Response: line 35 - reference 3 was moved to line 35

Line 37 – pls write ‘ most of the energy’

Response: It is corrected. Thank you.

Line 39 – pls remove ‘(2016)’. Also remove from other places too since you are using numbers as ref.

Response: It is corrected. Thank you.

Line 55 – pls remove one ‘E’ from experimental.

Response: It is corrected. Thank you.

Line 77 – define CO2 at first mention.

Response: It is corrected. Thank you.

In the study [] is too much in the write up and it affects the writing flow. Kindly read published papers to fix this.

Line 78 – replace ‘increase’ with concentration and ‘harder’ with ‘intense’

Response: It is corrected. Thank you.

Line 81 – pls replace indoor air quality with the abbreviation (IAQ) you already defined. Do same for others.

Response: It is corrected. Thank you.

Line 83 to 84 – pls write ‘which can be a result of’ as SBS cause(s) are sometimes unknown

Response: It is corrected. Thank you.

Line 87 – remove ‘WHO 1988’, else you are quoting twice

Response: It is corrected. Thank you.

Line 89 and 90 – pls rewrite the sentence

Response: The sentence is rephrased.

Line 109 and 110 – pls rephrase the sentence

Response: The sentence is rephrased.

Line 116 – write ‘with the aim’

Response: It is corrected. Thank you.

Line 123 – write ‘this paper’

Response: It is corrected. Thank you.

Line 126 to 128 – pls rephrase

Response: The sentence is rephrased.

Line 130 – pls replace installation with installing

Response: It is corrected. Thank you.

Line 133 – start with ‘The studied building is listed as a historical building ……’

Response: It is corrected. Thank you.

Line 162 – write ‘dusts are’

Response: We do not understand if you think to rewrite „duct“ to „dusts“.

Line 163 – ‘Air supply and exhaust are’

Response: It is corrected. Thank you.

Line 165 – you already explained this in the previous sentence

Response: The sentence is removed.

Kindly explain wat you mean by conservationist or remove the sentence about this

Response: We removed the two sentences about this.

Line 175 – pls rephrase the sentence

Response: The sentence is rephrased.

Label fig 4 and 5 for clarity

Response: Figures are improved.

Line 185 – write ‘the investigation of HVAC system for energy…….’

Response: It is corrected. Thank you.

Line 186 – replace the comma with ‘and’

Response: It is corrected. Thank you.

Line 189 – write ‘were not done’

Response: It is corrected. Thank you.

Line 192 – write ‘through’ and not throughout

Response: It is corrected. Thank you.

Fig 6. Should be made brighter

Response: Figure is improved.

Fig 6. Was measurement only done in room A?

Response: Yes, the measurement was done in classroom A.

Line 198. Write ‘with the windows and doors closed’

Response: It is corrected. Thank you.

Line 200. ‘up to the beginning’

Response: It is corrected. Thank you.

Line 213. Kindly put the correct sign for quotation for ‘breathing zone’

Response: It is corrected. Thank you.

Line 217. Pls remove ‘only’

Response: It is corrected. Thank you.

Line 218. Just write Table 1 shows the scales of the subjective evaluation

Response: It is corrected. Thank you.

Line 226. Write ‘the determined parameters are presented as’

Response: It is corrected. Thank you.

Line 228. What is ‘loss by heat recovery ventilation’?

Response: Information are rewritten - lines 248-259.

Line 233. Is it the measured temp. or average temp?

Response. It is average temperature.

This is not clear from the write up and the table. Can you explain why you have average RH and average outdoor temp and the others are not average?

Response: Yes, we corrected these parameters.

Fig 7. Correct the spelling for ‘consumption’.

Response: It is corrected. Thank you.

Line 241. Pls rewrite the sentence

Response: The sentence is rephrased.

Line 245. Pls rewrite ‘which is needed to prevent of water vapour condense’

Response: The sentence is rephrased.

The four categories mentioned in the fig (fig 7) and those mentioned from line 241 to 247 differs. E,g., electric energy consumption for fans.

Response: It is corrected. Thank you.

Line 249. Pls rewrite the sentence ’The average outdoor temp. in borders of…’ do same for the sentence that starts with ‘The reason was that’

Response: It is corrected. Thank you.

Line 255. Pls remove ‘is’

Response: It is removed. Thank you.

Line 258. Pls rewrite the sentence

Response: It is rewritten. Thank you.

Line 262. Pls remove ‘more significant’ except you want to explain how it was more significant.

Response: It is removed. Thank you.

Line 263 and 264. Pls remove 28% or write the % for PM concs. Too

Response: It is removed. Thank you.

Line 273. How did you link air quality with co2 conc? How did you know that odour was caused by VOCs?

Response: We removed this sentence.

Line 283. Pls write PM10 well and do same at other places too.

Response: It is corrected. Thank you.

You did not discuss why PM was low in some studies and why it was high in others. What can be done to make it stay within recommendation?

Response: The discussion is added – lines 290-293, 295-298, 399-305

Line 295 put ref. for permissible noise level.

Response: Reference is added.

Line 297. Pls put the ref. in the reference list.

Response: Reference is added.

Line 298. You evaluated lighting and not lightning.

Response: It is corrected. Thank you.

Line 298 to 302. Pls rearrange the sentences. Put the sentence that starts with ‘Due’ first, ‘Spears et al.’ second and then ‘lighting’

Response: The sentences are rearranged.

Line 308. Pls write ‘movement’

Response: It is corrected. Thank you.

Line 325. Pls rewrite

Response: It is rewritten. Thank you.

Line 317 to 345. Pls discuss what the correlations depicts/show

Response: Description of correlations is added – lines 359-366

Line 347. Pls rewrite ‘only one correspondent was smoker’

Response: It is corrected. Thank you.

Line 348. Pls remove ‘The’

Response: It is removed. Thank you.

Very weak discussion of results. Limiting factors were also not discussed.

Response: Discussion is added – lines 383-390

Limiting factors are added – lines 368-369.

Line 360. Pls remove ‘investigate’ and put ‘a’

Response: It is corrected. Thank you.

Line 362. You did not study chemical factors.

Response: The term chemical factors are removed. Thank you.

Line 366. Pls remove ‘according to Pettenkofer’

Response: It is removed. Thank you.

Line 367 and 368. Pls remove ‘performed using statistica software’

Response: It is removed. Thank you.

Line 369 to 379 should be in the discussion section

Response: We moved this section to the Results and Discussion chapter.

Reviewer 2 Report

Please replace "from" to "for" in the title. 

Also, explain a bit on Full-scale computer model (what details will be included and what aspect will be further explored) in the conclusion section.

Author Response

Comments and Suggestions for Authors

Please replace "from" to "for" in the title.

Response: Is is corrected. Thank you.

Also, explain a bit on Full-scale computer model (what details will be included and what aspect will be further explored) in the conclusion section.

Response: Information are added - lines 402-405. Thank you.

Round 3

Reviewer 1 Report

The authors should take their time to properly present the results of their study. They should ask the editor for more time. The paper as currently written cannot be published as there are so many grammatical mistakes in it. In addition, authors should painstakingly read published papers especially those published with IJERPH. It is only after this that a good review can be done.

Below are few examples from the introduction section of some of the many mistakes in the paper. They are many as from the first and second reviews. There are more in other section(s) of the paper.

line 17. abbreviate IEQ at first mention and not at second mention.

line 32. put comma after HVAC and remove 'the'

line 38. change suggests to suggested

line 99. write 'indicated'

line 107. write 'in another study'

line 114. remove 'in' ag´after et al.

line 118. put 'when' after even

line 125. remove 'in' after et al-

line 126. remove 'the'

line 130. pls rewrite the sentence that started with 'Due'

line 132. remove 'the'. Explain IEQ at first mention.

line 136. write IAQ

line 143 write IEQ

line 211. remove 'to provide'

line 216. rewrite the sentence that starts with 'Results'

line 218 to 219 should be in the discussion section.

please note that the above is just few of the mistakes seen (introduction and abstract). As said earlier, kindly take your time to properly present your results for publication.

Author Response

The authors thanks to reviewer for valuable notes.

We added references from IJERPH - lines 293, 294, 374, 375, 402-405.

The manuscript was given for English editing.
